# Cardiologist-Performed Baseline Evaluation with an Assessment of Coronary Status for Prostate Cancer Patients Undergoing Androgen Deprivation Therapy: Impact on Newly Diagnosed Coronary Artery Disease

**DOI:** 10.3390/cancers15164157

**Published:** 2023-08-17

**Authors:** Maximilien Rogé, Valentine Guimas, Emmanuel Rio, Loïg Vaugier, Tanguy Perennec, Joachim Alexandre, Stéphane Supiot, Elvire Martin Mervoyer

**Affiliations:** 1Department of Radiation Oncology, Centre Henri Becquerel, 1 Rue d’Amiens, 76000 Rouen, France; 2Department of Radiation Oncology, Institut de Cancérologie de l’Ouest, Bd. Professeur Jacques Monod, 44800 Saint-Herblain, France; 3INSERM U1086 ANTICIPE, Normandie Medecine University, UNICAEN, Avenue du Général Harris, 14000 Caen, France; 4Department of Pharmacology, PICARO Cardio-Oncology Program, CHU de Caen-Normandie, Avenue de la Côte de Nacre, 14000 Caen, France; 5Department of Cardio-Oncology, Institut de Cancérologie de l’Ouest, Bd. Professeur Jacques Monod, 44800 Saint-Herblain, France

**Keywords:** prostate cancer, androgen deprivation therapy, stress test, coronary artery disease, cardiac risk factor, cardiac screening, cardiac workup, cardio-oncology

## Abstract

**Simple Summary:**

This retrospective study investigated newly diagnosed coronary artery disease (CAD) in prostate cancer patients treated with Androgen Deprivation Therapy and who underwent cardio-onco evaluation at a Cancer Institute. The cardio-onco evaluation included a systematic evaluation of coronary status. Among the 34 patients evaluated, 20.6% were diagnosed with CAD, often asymptomatic with a median diagnosis time of 5 months. Of the 27 patients without CAD, 44.4% underwent a therapeutic intervention by the cardiologist. Our results underscore of CAD screening in this population; however, further research is needed to determine whether routine CAD screening for these patients would result in significant clinical benefits.

**Abstract:**

Background: Given the potential cardiovascular risks of androgen deprivation therapy (ADT), it is essential to identify patients who may be at an increased risk for coronary artery disease (CAD). Despite the recent ESC recommendations, there is no consensus on when to refer a patient to a cardiologist for further evaluation. Objective: To report on new diagnoses of CAD in patients with prostate cancer (PCa) requiring ADT who underwent a systematic cardio-onco evaluation with an assessment of their coronary status. Design, Setting, and Participants: This is a retrospective, monocentric study that included patients with PCa who had completed a cardio-onco evaluation with an assessment of their coronary status in the cardio-oncology department at the Western Cancer Institute, Nantes, between January 2019 and August 2022. Intervention: The baseline cardio-onco evaluation included a physical exam, transthoracic echography, and electrocardiogram, followed with a systematic evaluation of their coronary status. Outcome Measurements and Statistical Analysis: The primary objective was to determine the incidence of newly diagnosed CAD. The secondary objective was to evaluate the number of changes in cardiovascular treatment. Results and Limitations: Among the 34 patients who underwent cardio-onco evaluation, 7 (20.6%) were diagnosed with CAD, with a median time to diagnosis of 5 months. Most patients were asymptomatic, with one who experienced a myocardial infarction. Of the 27 patients without CAD, 44.4% underwent a therapeutic intervention by the cardiologist, with no cardiac deaths during follow-up. Overall, 55.9% of patients had a therapeutic intervention after the cardio-onco evaluation. Conclusions: The high incidence of newly diagnosed CAD in asymptomatic patients supports the need for screening for CAD in this population. Further research is needed to determine whether routine screening for CAD in patients receiving ADT would result in significant clinical benefits.

## 1. Introduction

Prostate cancer (PCa) is the second most commonly diagnosed cancer and the sixth leading cause of cancer death among men worldwide, with an estimated 1,276,000 new cancer cases and 359,000 deaths occurring in 2018 [1]. A registry study conducted among men with PCa in Sweden and the United States found that cardiovascular disease, specifically ischemic heart disease, was the second most common cause of death after PCa [2].

Androgen deprivation therapy (ADT) is the cornerstone in the current systemic treatment of PCa for localized or metastatic disease [3,4]. However, it can lead to various metabolic changes that increase the risk of cardiovascular disease, such as increased fat mass [5], changed lipid profiles [6], and increased insulin resistance, which could result in diabetes [7] and is associated with dyslipidemia, which could result in metabolic syndrome [8].

Although there are some conflicting data [9,10,11,12], the American Heart Association, American Cancer Society, and American Urological Association jointly released a science advisory on the increased cardiovascular risks of ADT, including coronary artery disease (CAD) [13].

Given the potential risks of ADT and the known link between PCa and well-known cardiovascular risk factors [14], such as age and obesity, it is crucial to identify patients who may be at an increased risk for CAD.

The European Society of Cardiology (ESC) recommends a baseline cardiovascular risk assessment and estimation of 10-year fatal and non-fatal CVD risk with SCORE2 or SCORE2-OP, followed by lifestyle risk control and patient education on healthy lifestyles prior to the initiation of androgen deprivation therapy [15].

However, there is still no consensus on when to refer a patient to a cardiologist for further evaluation. In this study, we aimed to investigate the incidence of newly diagnosed CAD in PCa patients who underwent a systematic baseline cardio-onco evaluation with an assessment of their coronary status by a cardiologist.

## 2. Material and Methods

### 2.1. Study Design and Participants

We used institutional registry databases to identify and extract data from consecutive men who had histologically confirmed PCa at the Western Cancer Institute, Nantes (ICO), and who had completed a baseline cardio-onco evaluation in the ICO cardio-oncology department between January 2019 and August 2022. Patients with PCa who required first- +/− second-generation ADT were systematically seen in the ICO cardio-oncology department if they did not already have a regular cardiologist.

Patients with metastatic castration-resistant PCa, those with a history of ADT treatment less than 1 year, and those who had already started ADT prior to the baseline cardio-onco evaluation were excluded from the study. 

When ADT was initiated after the baseline cardio-onco evaluation, it was planned for a duration of 6 months or 24 to 36 months for localized prostate cancers, and lifelong for metastatic prostate cancers.

Patients who did not have a complete cardio-onco evaluation were also excluded. The complete cardio-onco evaluation included the baseline cardio-onco evaluation with a physical exam, transthoracic echography (TTE), and electrocardiogram (ECG), followed by at least one complementary cardiac examination to assess coronary status (Stress echocardiography, CCTA: coronary computed tomography angiography, non-contrast cardiac CT for calcium scoring, cardiac scintigraphy, and coronary angiography). 

The study was approved by the Institutional Ethics Committee of Angers Hospital in France (protocol code 2022-159, date of approval 2 November 2022).

### 2.2. Procedures

We collected the initial characteristics of the PCa (Initial PSA, Gleason, ISUP, TNM) and classified the localized PCa according to the d’Amico classification [16]. For patients with mCSPC, we defined high-risk patients according to the LATITUDE definition (≥2 high-risk features of: ≥3 bone metastasis, visceral metastasis, or Gleason ≥ 8) and high-tumoral-volume patients according to the CHAARTED definition (≥4 bone metastasis, including ≥ 1 outside vertebral column or pelvis and/or visceral metastasis) [17,18]. We also recorded if patients had synchronous or metachronous metastatic disease.

We collected the patients’ cardiovascular history, risk factors, and the results of their clinical, biological (lipid profile, GFR: glomerular filtration rate, and glycaemia), and complementary cardiac examinations. The QT interval was corrected according to Fridericia’s formula.

For patients with available data (age, systolic blood pressure, smoking status, diabetes, and lipid profile), we determined at the date of the baseline cardio-onco evaluation their Systematic Coronary Risk Estimation 2 (SCORE2) (<70 years old) or SCORE2-OP (Older Persons) (>70 years old) in accordance with ESC recommendations [15,19,20]. The patients (50–69 years) were stratified into low risk (SCORE2 < 5%), moderate risk (SCORE2: 5–10%), and high risk (SCORE2 > 10%). Older patients (≥70 years) were also stratified into low risk (SCORE2-OP < 7.5%), moderate risk (SCORE2-OP: 7.5–15%), and high risk (SCORE2-OP > 15%).

We defined according to the ESC guidelines [21] a “History of cardiovascular disease” as a history of myocardial infarction, angina, coronary stent, peripheral vascular disease, coronary artery bypass graft, stroke, or transient ischemic attack; patients as having “Cardiovascular Risks” if they had any of the following criteria, such as: arterial hypertension, dyslipidemia, diabetes, ex or current smoker, or a family cardiovascular disease history; and “Documented coronary artery disease (CAD)” as a condition diagnosed with CCTA or coronary angiography that required intervention (medical treatment, stent, or coronary artery bypass graft). Patients did not need to have any cardiac symptoms or events to satisfy this criterion.

We also recorded if the cardiologist prescribed cardiovascular medications (new treatment or modification). 

The follow-ups were calculated from the date of the baseline cardio-onco evaluation to the date of the last news.

### 2.3. Statistical Analysis

The results for the categorical variables are expressed as absolute numbers and percentages. The results for the continuous variables are expressed as medians (interquartile ranges). The available data are specified for each variable. No statistical test was performed. The analyses were performed using the R 4.2.2 software. 

### 2.4. Outcomes

The primary objective of this study was to determine the incidence of newly diagnosed CAD in PCa patients who underwent a systematic baseline cardio-onco evaluation with an assessment of their coronary status by a cardiologist prior to starting androgen deprivation therapy (ADT). The secondary objective was to evaluate the number of cardiovascular risk management treatment introductions or changes made by the cardiologist during the baseline evaluation.

## 3. Results

### 3.1. Population

Between January 2019 and August 2022, 103 patients with PCa were seen in the ICO cardio-oncology department. After applying our inclusion and exclusion criteria, we included 34 patients in our study (Appendix A).

Table 1 summarizes their oncological characteristics. The median age at diagnosis was 65.0 (50–79) years. In total, 31 patients had localized PCa, with 3 favorable intermediate, 10 unfavorable intermediate, and 18 high risks. Three patients had low-risk and low-volume metachronous metastatic disease with five or fewer metastases. At the time of the baseline cardio-onco evaluation, twenty patients (58.8%) had a prior prostatectomy and nine (26.5%) patients had received a prior prostate or prostate bed radiotherapy.

### 3.2. Baseline Cardio-Onco Evaluation

The cardiovascular characteristics at the baseline cardio-onco evaluation are summarized in Table 2. The median age at this evaluation was 71.0 (52–81) years old. Only 2 patients (5.9%) had a history of cardiovascular disease, but most patients had cardiovascular risk factors (*n* = 29, 85.3%). Eleven patients (32.4%) were obese (BMI ≥ 30 kg/m^2^) and eleven patients (32.4%) were overweight (BMI: [25–30] kg/m^2^). According to the SCORE2-OP calculation, and for whom data were available, all the patients (≥70 years) had a 10-year cardiovascular moderate or high risk (greater than 7.5%), with 4 patients (n = 4/11, 36.4%) having a high risk >15%. However, according to the SCORE2 calculation (<70 years), only a few patients with available data had a 10-year cardiovascular high risk (>10%) (n = 2/12, 8.3%).

At the baseline cardio-onco evaluation, according to our inclusion criteria, all the patients had a physical exam, TTE, and ECG with at least one complementary cardiac exploration prescribed by the cardiologist. The main results of the TTE and ECG are available in Table 3. Of the complementary cardiac exploration prescribed by the cardiologist, and after a median duration of 58 days (0–208), 27 (79.4%) patients underwent a cardiac stress test (25 (73.5%) stress echocardiography and 2 (5.9%) cardiac scintigraphy), 4 patients (11.8%) a CCTA, and 3 patients (8.8%) CT coronary calcium scoring.

The details of the 27 cardiac stress tests performed are available in the (Appendix A). None of these tests were clinically positive (no symptoms were reported during the test). Of the four patients (14.8%) who experienced ECG rhythm disorder, two (50.0%) had ventricular and/or supraventricular excitability and two (50.0%) had ischemia signs. Dyskinetic segments were detected by stress in nine patients (33.3%). 

### 3.3. Follow-Up

After a median follow-up of 9 months (1–36), all the localized PCa patients without previous prostate or prostate bed radiotherapy received radiation treatment concurrent with ADT. Seven cases of CAD were diagnosed (20.6%). The median time to diagnose CAD was 5 months (2–7). Six out of seven cases were diagnosed in asymptomatic patients. The last one was diagnosed after the patient experienced a myocardial infarction 2 months after a positive stress echocardiography and before the completion of CCTA (Table 4). None of these patients had a history of cardiovascular disease, but five (71.4%) had cardiovascular risk factors. The CAD was treated with medical treatment alone for four patients (57.1%) and with stenting + medical treatment for three patients (42.9%). 

Among the 27 patients without a diagnosis of CAD (79.4%), 12 of them (44.4%) underwent a therapeutic intervention by the cardiologist. One diabetic patient had a diagnosis of microcirculation disorder that required beta-blockers. Four patients required an introduction or modification of cardiovascular medications for hypertension, five patients for dyslipidaemia, and two patients for both. No patients died of cardiac causes during follow-up. 

Overall, of the 34 patients who underwent a systematic cardiac workup, 19 (55.9%) had a therapeutic intervention after this evaluation.

## 4. Discussion

This study is the first to report systematic cardio-onco evaluation results, including coronary status, in a real-life patient population treated with ADT. After a median follow-up of 9 months, seven (20.6%) patients were diagnosed with CAD, including one (2.9%). who experienced a myocardial infarction.

In the PRONOUNCE trial [22], which compared the cardiovascular safety of a GnRH agonist and antagonist, a cardiologist ensured that baseline secondary prevention medications for atherosclerotic cardiovascular disease (ASCVD) were optimized according to the guideline recommendations, but systematic coronary evaluation were not performed. Five hundred and forty-five patients were included, with 86.8% having pre-existing cardiac disorders. At one year, the authors reported a total of eight myocardial infarctions (1.5%). Wallach et al., in a retrospective, propensity-matched cohort study, included 2226 patients, with 52.7% of them having pre-existent CAD. At 12 months, 35 patients experienced a myocardial infarction (1.6%) [23]. In our study, with a shorter follow-up and a small number of patients, only one patient experienced a myocardial infarction, which is consistent with previous studies and real-life clinical practice. However, 85.3% of the patients included in our study had cardiovascular risk factors, which explains the high proportion of asymptomatic CAD (n = 6, 17.7%) diagnosed through a systematic coronary evaluation.

In the management of patients treated with androgen deprivation therapy (ADT) for PCa, different scientific societies have issued recommendations with varying degrees of specificity regarding cardiac evaluation. 

The European Society of Cardiology (ESC), in 2022, recommended before androgen deprivation therapy a baseline cardiac risk evaluation with SCORE2, ECG, patient education on healthy lifestyles, and lifestyle risk factor control. Nevertheless, a systematic evaluation of coronary status is not recommended [15]. The American Heart Association, American Cancer Society, and American Urological Association published recommendations in 2010 and proposed an initial and yearly evaluation with assessments of blood pressure, lipid profile, and glucose level. The authors also stated that patients who are thought to benefit from ADT should not be referred systematically to cardiologists for evaluation before the initiation of ADT. Even for patients referred to a cardiologist, it was not recommended that the cardiologist perform a specific test or coronary intervention before starting ADT [13].

In contrast, in 2022, the European Association of Urology (EAU) recommended that all patients receiving ADT should undergo screening for diabetes and a blood lipid level assessment, and advised that men with a history of cardiovascular disease and men older than 65 years should receive a cardiology consultation prior to initiating ADT. The Canadian Urological Association published their guidelines in 2021, which recommended lifestyle modifications, a baseline physical examination, and laboratory investigations on factors such as fasting plasma glucose and lipid profile. In addition, the authors recommended that patients with a history of myocardial infarction (MI) or stroke should be referred to a cardiologist or cardio-oncologist for assessment and medical optimization at the time of initiating ADT [24]. Australian urologists published their own recommendations in 2014. They proposed the screening and management of individual cardiovascular risk factors and to refer patients with a history of myocardial infarction, history of cerebrovascular events, congestive heart failure, or peripheral arterial disease to a cardiologist [25].

Overall, there is still heterogeneity among the recommendations regarding cardiac evaluation for patients undergoing ADT, with different societies suggesting varying levels of screening and referrals to specialists.

Another goal for cardio-oncologists after preventing cardiotoxicity is also to identify high-risk patients for which the ADT choice could be relevant. Some studies have suggested that GnRH antagonists are associated with a significantly lower overall mortality and cardiovascular events compared to agonists [26]. However, the PRONOUCE trial and a large retrospective study (n = 7800) published by Wallach et al. did not find a cardiovascular safety difference between these two treatments in patients with a history of cardiovascular disease [22,23]. The choice of new ADT generation can also help mitigate cardiovascular risk in the highest-risk patients. For instance, abiraterone appears to be more strongly associated with an increased risk of cardiac events compared to enzalutamide [27]. Lastly, a new oral GnRH antagonist (relugolix) was evaluated in a phase 3 trial involving men with advanced PCa and showed promising results, with a 54% lower risk of major adverse cardiovascular events compared to agonist GnRH [28].

To optimize the timing of visits to cardio-oncologists and the performance of complementary cardiac tests without having to delay ADT initiation, there is a need to better select the patients who require a cardiac evaluation, given the low number of dedicated services and increasing demand [15]. The authors of the ESC guidelines recommend the use of SCORE2 or SCORE2-OP to stratify the cardiovascular risks in patients receiving ADT without a history of CVD, as no cardiovascular risk calculator has been developed for these patients [15]. However, our study showed that SCORE2 alone may not be sufficient to determine if a cardiac assessment and coronary evaluation is necessary, as one patient with newly diagnosed CAD was considered as low risk according to SCORE2. Therefore, including the age factor and cardiovascular disease history (as recommended by other societies) into the decision-making process may be a good option for determining which patients should be referred to a cardiologist and undergo a coronary evaluation.

It is important to note that our retrospective study has inherent limitations, such as a possible selection bias. The heterogeneity in the type and sequence of the additional cardiac tests performed in this study may also affect the reliability of the results. Additionally, the high rate of missing data for SCORE2 may limit the generalizability of the findings. Furthermore, the study could not demonstrate whether the diagnosis and treatment of asymptomatic CAD led to improved clinical outcomes in this specific patient population. This is an important limitation, as it is unclear whether routine screening for CAD in patients receiving ADT would result in significant clinical benefits. 

A significant limitation of this study is its small sample size and the lack of baseline risk stratification into low- and high-risk cardiovascular risk for these patients prior to the initiation of ADT. Therefore, this study’s results may be utilized to propose a pathway for larger studies, but we should be very careful in generalizing the results obtained from the observations of a small-sample-size study.

## 5. Conclusions

Different scientific societies recommend baseline cardiac risk evaluation, patient education on healthy lifestyles, lifestyle risk factor control, and screening for cardiovascular risk factors.

Our study highlights the value of systematic baseline cardio-onco evaluation, including an assessment of coronary status by a cardiologist in PCa patients treated with ADT. 

However, our study has limitations and it remains unclear how to determine which patients should undergo a systematic cardiac oncology evaluation and coronary assessment. Further research is needed to determine whether a systematic cardiac oncology evaluation with a coronary assessment would result in significant clinical benefits.

## Figures and Tables

**Table 1 cancers-15-04157-t001:** Initial oncological characteristics.

	Localized	Metastatic	Total
	(N = 31)	(N = 3)	(N = 34)
**Age at diagnosis**			
Median (Min, Max)	65.0 [50.0, 79.0]	58.0 [58.0, 65.0]	65.0 [50.0, 79.0]
Missing	1 (3.2%)	0 (0%)	1 (2.9%)
**WHO performance status**			
0	38 (86.4%)	14 (87.5%)	52 (86.7%)
1	5 (11.4%)	1 (6.3%)	6 (10.0%)
**Initial PSA**			
Mean (SD)	15.7 (15.1)	12.5 (6.50)	15.4 (14.5)
Median (Min, Max)	7.95 [4.50, 64.0]	9.46 [8.14, 20.0]	8.40 [4.50, 64.0]
Missing	1 (3.2%)	0 (0%)	1 (2.9%)
**ISUP**			
1	1 (3.2%)	0 (0%)	1 (2.9%)
2	6 (19.4%)	1 (33.3%)	7 (20.6%)
3	12 (38.7%)	2 (66.7%)	14 (41.2%)
4	12 (38.7%)	0 (0%)	12 (35.3%)
5	0 (0%)	0 (0%)	0 (0%)
Missing	0 (0%)	0 (0%)	0 (0%)
**Amico**			
Low risk	0 (0%)	0 (0%)	0 (0%)
Favorable Intermediate risk	3 (9.7%)	0 (0%)	3 (8.8%)
Unfavorable Intermediate risk	10 (32.3%)	0 (0%)	10 (29.4%)
High risk	18 (58.1%)	0 (0%)	18 (52.9%)
Metastatic	0 (0%)	3 (100%)	3 (8.8%)
**cT**			
T1c	4 (12.9%)	2 (66.7%)	6 (17.6%)
T2	17 (54.8%)	0 (0%)	17 (50.0%)
T3a	5 (16.1%)	0 (0%)	5 (14.7%)
T3b	2 (6.5%)	0 (0%)	2 (5.9%)
Tx	3 (9.7%)	1 (33.3%)	4 (11.8%)
**cN**			
N0	26 (83.9%)	0 (0%)	26 (76.5%)
N1	4 (12.9%)	3 (100%)	7 (20.6%)
Nx	1 (3.2%)	0 (0%)	1 (2.9%)
**pT**			
pT2	7 (22.6%)	0 (0%)	7 (20.6%)
pT3a	6 (19.4%)	0 (0%)	6 (17.6%)
pT3b	5 (16.1%)	2 (66.7%)	7 (20.6%)
No surgery	13 (41.9%)	1 (33.3%)	14 (41.2%)
**pN**			
pN0	15 (48.4%)	1 (33.3%)	16 (47.1%)
pN1	2 (6.5%)	1 (33.3%)	3 (8.8%)
pNx	1 (3.2%)	0 (0%)	1 (2.9%)
No surgery	13 (41.9%)	1 (33.3%)	14 (41.2%)
**cM**			
0	31 (100%)	0 (0%)	31 (91.2%)
1a	0 (0%)	2 (66.7%)	2 (5.9%)
1b	0 (0%)	1 (33.3%)	1 (2.9%)
**Number of metastases**			
0	31 (100%)	0 (0%)	31 (91.2%)
1–5	0 (0%)	3 (100%)	3 (8.8%)
>5	0 (0%)	0 (0%)	0 (0%)
**M+ disease**			
Non metastatic	31 (100%)	0 (0%)	31 (91.2%)
Synchronous	0 (0%)	0 (0%)	0 (0%)
Metachronous	0 (0%)	3 (100%)	3 (8.8%)
**Imaging for M+ diagnosis**			
Non metastatic	31 (100%)	0 (0%)	31 (91.2%)
Conventional Imaging	0 (0%)	0 (0%)	0 (0%)
Metabolic Imaging	0 (0%)	3 (100%)	3 (8.8%)
**Risk**			
Non metastatic	31 (100%)	0 (0%)	31 (91.2%)
Low risk	0 (0%)	3 (100%)	3 (8.8%)
High risk	0 (0%)	0 (0%)	0 (0%)
**Tumoral Volume**			
Non metastatic	31 (100%)	0 (0%)	31 (91.2%)
Low volume	0 (0%)	3 (100%)	3 (8.8%)
High volume	0 (0%)	0 (0%)	0 (0%)

PSA: Prostate Specific Antigen; WHO: World Health Organization; and M+: Metastatic.

**Table 2 cancers-15-04157-t002:** Cardiovascular characteristics at the baseline cardio-onco evaluation.

	Localized	Metastatic	Total
	(N = 31)	(N = 3)	(N = 34)
**Age**			
Median (Min, Max)	70.0 [52.0, 81.0]	72.0 [68.0, 74.0]	71.0 [52.0, 81.0]
**History of ADT**			
No	25 (80.6%)	1 (33.3%)	26 (76.5%)
Yes	6 (19.4%)	2 (66.7%)	8 (23.5%)
**History of Cardiovascular Disease ^a^**			
No	29 (93.5%)	3 (100%)	32 (94.1%)
Yes	2 (6.5%)	0 (0%)	2 (5.9%)
**Cardiovascular Risk factors ^b^**			
No	5 (16.1%)	0 (0%)	5 (14.7%)
Yes	26 (83.9%)	3 (100%)	29 (85.3%)
**Tobacco**			
Non-smoker	11 (35.5%)	0 (0%)	11 (32.4%)
Current or ex-smoker	20 (64.5%)	3 (100%)	23 (67.6%)
**Pack years (PA)**			
<15	6 (19.4%)	1 (33.3%)	7 (20.6%)
15–30	3 (9.7%)	0 (0%)	3 (8.8%)
30–45	2 (6.5%)	1 (33.3%)	3 (8.8%)
>45	0 (0%)	1 (33.3%)	1 (2.9%)
Missing	10 (32.3%)	0 (0%)	10 (29.4%)
**Arterial Hypertension**			
No	20 (64.5%)	0 (0%)	20 (58.8%)
Yes	11 (35.5%)	3 (100%)	14 (41.2%)
**Diabetes**			
No diabetes	28 (90.3%)	3 (100%)	31 (91.2%)
Type 1	2 (6.5%)	0 (0%)	2 (5.9%)
Type 2	1 (3.2%)	0 (0%)	1 (2.9%)
**Dyslipidemia**			
No	24 (77.4%)	3 (100%)	27 (79.4%)
Yes	7 (22.6%)	0 (0%)	7 (20.6%)
**Renal insufficiency**			
Yes (<60 mL/min)	5 (16.1%)	0 (0%)	5 (14.7%)
No (>60 mL/min)	21 (67.7%)	0 (0%)	21 (61.8%)
Missing	5 (16.1%)	3 (100%)	8 (23.5%)
**Familial CV Disease**			
No	26 (83.9%)	3 (100%)	29 (85.3%)
Yes	5 (16.1%)	0 (0%)	5 (14.7%)
**BMI (kg/m^2^)**			
18.5–20	0 (0%)	0 (0%)	0 (0%)
20–25	11 (35.5%)	1 (33.3%)	12 (35.3%)
25–30	11 (35.5%)	0 (0%)	11 (32.4%)
30–35	8 (25.8%)	2 (66.7%)	10 (29.4%)
35–40	1 (3.2%)	0 (0%)	1 (2.9%)
**SCORE2 (50–69 y)**			
<5% (low risk)	3 (9.7%)	0 (0%)	3 (8.8%)
5–10% (moderate risk)	7 (22.6%)	0 (0%)	7 (20.6%)
10–15% (high risk)	2 (6.5%)	0 (0%)	2 (5.9%)
Missing	3 (9.7%)	1 (33.3%)	4 (11.8%)
**SCORE2-OP (≥70 y)**			
<7.5% (low risk)	0 (0%)	0 (0%)	0 (0%)
7.5–15% (moderate risk)	7 (22.6%)	0 (0%)	7 (20.6%)
15–20% (high risk)	3 (9.7%)	0 (0%)	3 (8.8%)
>20% (high risk)	1 (3.2%)	0 (0%)	1 (2.9%)
Missing	4 (12.9%)	2 (66.7%)	6 (17.6%)

^a^ History of cardiovascular history corresponds to myocardial infarction, angina, coronary stent, peripheral vascular disease, coronary artery bypass graft, stroke, or transient ischemic attack. ^b^ Cardiovascular risks correspond to arterial hypertension, dyslipidemia, diabetes, current or ex-smoker, or familial cardiovascular Disease. ADT: Androgen Deprivation Therapy; and BMI: Body Mass Index.

**Table 3 cancers-15-04157-t003:** Results of baseline cardiac workup.

	Localized	Metastatic	Total
	(N = 31)	(N = 3)	(N = 34)
**Rhythm**			
Normal Sinus Rhythm	30 (96.8%)	3 (100%)	33 (97.1%)
Atrial Fibrillation	0 (0%)	0 (0%)	0 (0%)
Missing	1 (3.2%)	0 (0%)	1 (2.9%)
**Cardiac Frequency**			
Mean (SD)	68.0 (12.6)	67.3 (9.07)	68.0 (12.2)
Missing	1 (3.2%)	0 (0%)	1 (2.9%)
**QRS**			
Normal	27 (87.1%)	3 (100%)	30 (88.2%)
Complete right BBB	1 (3.2%)	0 (0%)	1 (2.9%)
Incomplete left BBB	2 (6.5%)	0 (0%)	2 (5.9%)
Complete left BBB	0 (0%)	0 (0%)	0 (0%)
Missing	1 (3.2%)	0 (0%)	1 (2.9%)
**QTc**			
Mean (SD)	414 (19.2)	399 (11.0)	413 (19.0)
Missing	1 (3.2%)	0 (0%)	1 (2.9%)
**LVEF**			
Median (Min, Max)	63.0 [30.0, 77.0]	70.0 [66.0, 70.0]	64.5 [30.0, 77.0]
Missing	2 (6.5%)	0 (0%)	2 (5.9%)
**Aortic valvulopathy**			
No	25 (80.6%)	2 (66.7%)	27 (79.4%)
Yes	3 (9.7%)	1 (33.3%)	4 (11.8%)
Missing	3 (9.7%)	0 (0%)	3 (8.8%)
**Mitral valvulopathy**			
No	26 (83.9%)	3 (100%)	29 (85.3%)
Yes	2 (6.5%)	0 (0%)	2 (5.9%)
Missing	3 (9.7%)	0 (0%)	3 (8.8%)
**Type of first complementary cardiac exploration**			
CCTA	3 (9.7%)	1 (33.3%)	4 (11.8%)
Stress Echocardiography	23 (74.2%)	2 (66.7%)	25 (73.5%)
Cardiac Scintigraphy	2 (6.5%)	0 (0%)	2 (5.9%)
CT Coronary calcium scoring	3 (9.7%)	0 (0%)	3 (8.8%)
**Time between baseline cardio-onco evaluation and first exploration (days)**			
Median (Min, Max)	59.0 [0, 208]	50.0 [15.0, 61.0]	58.0 [0, 208]

BBB: Bundle Branch Block; LVEF: Left Ventricular Ejection Fraction; CCTA: Coronary computed tomography angiography; and CT: computed tomography.

**Table 4 cancers-15-04157-t004:** Details of patients with diagnosed coronary artery disease.

	Total
	(N = 7)
**Age**	
Median (Min, Max)	66.0 [63.0, 75.0]
**History of Cardiovascular Disease ^a^**	
No	7 (100%)
**Cardiovascular Risk Factors ^b^**	
No	2 (28.6%)
Yes	5 (71.4%)
**SCORE2 or SCORE2-OP**	
Low risk	1 (14.3%)
Moderate risk	4 (57.1%)
High risk	1 (14.3%)
Missing	1 (14.3%)
**Exams performed to diagnose CAD**	
CCTA	1 (14.3%)
CCTA → Stress TTE → ICA	1 (14.3%)
Stress TTE → ICA	1 (14.3%)
Stress TTE → CCTA	2 (28.6%)
Stress TTE → CCTA → ICA	1 (14.3%)
Stress TTE → Calcium scoring → ICA	1 (14.3%)
**CAD Treatment**	
Stenting + medical treatment	3 (42.9%)
Medical treatment alone	4 (57.1%)

^a^ History of cardiovascular history corresponds to myocardial infarction, angina, coronary stent, peripheral vascular disease, coronary artery bypass graft, stroke, or transient ischemic attack. ^b^ Cardiovascular risks correspond to arterial hypertension, dyslipidemia, diabetes, current or ex-smoker, or familial cardiovascular disease. CCTA: Coronary computed tomography angiography; TTE: Transthoracic echography; and ICA: Invasive coronary angiography.

## Data Availability

The data presented in this study are available upon request from the corresponding author. The data are not publicly available for confidentiality reasons.

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
