# Peer review of "Cardiologist-Performed Baseline Evaluation with an Assessment of Coronary Status for Prostate Cancer Patients Undergoing Androgen Deprivation Therapy: Impact on Newly Diagnosed Coronary Artery Disease"

_cancers, 2023, doi:10.3390/cancers15164157_

Round 1

Reviewer 1 Report

Comments on study design:

Methods: Why patients seen by a regular cardiologist were excluded? This may introduce referral bias and reduce ability to look at incidence of CAD in ADT patients at your institution. Unclear of how many patients are seen by general cardiology versus cardio oncology. Entering all patients evaluated by any cardiologist may improve your sample size and then you can try to compare between general cardiologist and cardio oncologist and establish if there is an added value in the cardio-oncology evaluation.

No statistical methods used. Just descriptive statistics. Any attempt to do a comparison by baseline risk to stratify low versus high risk? A limitation is clearly the limited number of patients (sample size) included in this study.

Discussion.

Discussion about heterogeneity on recommendations or cardiovascular screening and management of prostate cancer patients treated with ADT. How does this study contribute to clarify this? What is the new information or insight provided by this small size database?

Probably a design where you can identify low risk and high-risk patients for CV complications prior to initiation of ADT treatment, follow them prospectively for CV complications, obtain data on the rate differential of CV events among these groups and then try to propose a CV screening/follow up pathway for prostate cancer and ADT based on their baseline risk, although this may require a multi-center collaboration.

Minor editing suggestions:

-Abstract: 1. Line 23. “it is still not consensual” may be easier to read with: “there is no consensus” 2. Line 32: Instead of “newly CAD” should read “newly diagnosed CAD”.  3. Line 33; Instead of “cardiovascular risk management treatment introductions or changes” should read “changes in cardiovascular treatment”

-Introduction: Lines 63-66:

“The European Society of Cardiology (ESC) recommends before androgen deprivation therapy a baseline cardiovascular risk assessment and estimation of 10-year fatal and 64 non-fatal CVD risk with SCORE2 or SCORE2-OP followed by lifestyle risk control and 65 patient education on healthy lifestyle (23)”

The European Society of Cardiology (ESC) recommends a baseline cardiovascular risk assessment and estimation of 10-year fatal and 64 non-fatal CVD risk with SCORE2 or SCORE2-OP followed by lifestyle risk control and 65 patient education on healthy lifestyle (23) prior to initiation of androgen depravation therapy.

Author Response

Methods: Why patients seen by a regular cardiologist were excluded? This may introduce referral bias and reduce ability to look at incidence of CAD in ADT patients at your institution. Unclear of how many patients are seen by general cardiology versus cardio oncology. Entering all patients evaluated by any cardiologist may improve your sample size and then you can try to compare between general cardiologist and cardio oncologist and establish if there is an added value in the cardio-oncology evaluation.

Patients already seen by a regular cardiologist are already being monitored and treated for cardiac conditions, including coronary artery disease (CAD). The purpose of our study is to demonstrate the incidence of CAD in a population of patients without prior cardiac follow-up, aiming to highlight the potential benefit of systematically referring certain patients with no cardiac history to a cardiologist.

No statistical methods used. Just descriptive statistics. Any attempt to do a comparison by baseline risk to stratify low versus high risk? A limitation is clearly the limited number of patients (sample size) included in this study.

You are absolutely right. This is a significant limitation of our study. Despite the relatively high percentage (20.6%) of diagnosed CAD in our study among asymptomatic patients without cardiovascular history, it only involves 7 out of the 34 patients, which does not allow statistical correlation and an identification of a statistically significant high-risk subgroup for CAD.

Discussion: Discussion about heterogeneity on recommendations or cardiovascular screening and management of prostate cancer patients treated with ADT. How does this study contribute to clarify this? What is the new information or insight provided by this small size database?

Different scientific societies do not recommend systematic cardiac evaluation or assessment of coronary status. Despite the limited number of patients, our study is original as it is the first to demonstrate that asymptomatic prostate cancer patients with no prior cardiac history are at risk of coronary artery disease (CAD), with 20.6% of patients in our study being affected.

This study initiates a crucial discussion, highlighting the imperative for prospective research to focus on three primary aspects: firstly, identifying the high-risk population susceptible to CAD (as SCORE2 does not appear sufficient for this population); secondly, establishing the most suitable cardiac screening protocol for these patients; and thirdly, demonstrating the clinical outcomes benefits of such strategies.

Probably a design where you can identify low risk and high-risk patients for CV complications prior to initiation of ADT treatment, follow them prospectively for CV complications, obtain data on the rate differential of CV events among these groups and then try to propose a CV screening/follow up pathway for prostate cancer and ADT based on their baseline risk, although this may require a multi-center collaboration.

You are absolutely right, and a design as you proposed would address the three previous points raised following your previous comment. Therefore, our study opens up a discussion, particularly concerning the necessity for comparative scientific data and recommendations on the cardiac screening protocol in this population.

About the minor revisions, you are right, thanks for your feedback We have modified the manuscript according to your review

Reviewer 2 Report

I appreciate the efforts you have put into this study and the valuable insights it provides on conducting a systematic cardiac oncology evaluation. I think the main limit is only about the low number of patients. However, I find your article to be well-written, and it offers valuable insights into the field of cardiac oncology evaluations for prostate cancer patients.

  1.  
  1.  

Author Response

Dear Reviewer,

Thanks a lot for your review and your comments.

Kind regards

Reviewer 3 Report

The manuscript by Roge et.al. retrospectively analyzed the baseline cardiovascular status of 34 patients with localized and metastatic prostate cancer undergoing ADT, and incidence of CAD within a median follow up of 9 months. Results in the study shed light on how to risk stratify patients for additional cardiac work up to prevent occurrence of CAD events following ADT.

Questions that need further clarification from authors

1. For patient with localized PC undergoing ADT, did they receive concurrent radiation treatment?

2. What is the ADT regimen used?

3. What factors determine which "complementary cardiac exam" the patient receives?

Author Response

  1. For patient with localized PC undergoing ADT, did they receive concurrent radiation treatment?

At the time of the baseline cardio-onco evaluation, 9 patients (26.5%) patients had a prior prostate or salvage radiotherapy.

At the time of the last follow-up, all localized prostate cancer patients without previous prostate or prostate bed radiotherapy received radiation treatment concurrent with ADT.

Since we are presenting the initial characteristics of the patients during the baseline cardio-onco evaluation (table 1 and table 2), we have included this information in the follow-up section.

  1. What is the ADT regimen used?

According to our inclusion criteria, ADT was not started before the baseline cardio-onco evaluation. When ADT was initiated after the baseline cardio-onco evaluation, it was planned for a duration of 6 months or 24 to 36 months for localized prostate cancers, and lifelong for metastatic prostate cancers.

We added this information in the Material and Methods section

  1. What factors determine which "complementary cardiac exam" the patient receives?

The baseline cardio-onco evaluation was standardized, but the choice of the complementary cardiac exam for coronary assessment was at the discretion of the cardiologist and based on individual patient factors, particularly considering that some patients were unable to undergo a maximal effort to evaluate their coronary status during a stress test.

Round 2

Reviewer 1 Report

In study limitations prior to the conclusions, I suggest that a phrase should be added with the following acknowledgment:

"A significant limitation of this study is the small sample size and the lack of baseline risk stratification in low and high-risk CV risk for these patients prior to initiation of ADT. Therefore, this study results may be utilized to propose a pathway for larger studies, but we should be very careful in generalizing the results obtained from the observations of a small sample size study".

Author Response

Thank you for your comment.

I have incorporated your suggested phrase just before the conclusion and removed the last sentence, which me seems redundant with your proposal, which originally read as follows: "While the findings of this study provide valuable insights, further research is necessary to confirm the results and assess the clinical usefulness of routine screening for CAD in patients receiving ADT. Additionally, future studies should focus on identifying which patients would benefit the most from routine screening."